# Facile Preparation of a Superhydrophobic iPP Microporous Membrane with Micron-Submicron Hierarchical Structures for Membrane Distillation

**DOI:** 10.3390/polym12040962

**Published:** 2020-04-20

**Authors:** Cuicui Hu, Zhensheng Yang, Qichao Sun, Zhihua Ni, Guofei Yan, Zhiying Wang

**Affiliations:** National-Local Joint Engineering Laboratory for Energy Conservation in Chemical Process Integration and Resources Utilization, School of Chemical Engineering and Technology, Hebei University of Technology, Tianjin 300000, China; hebuthcc2017@163.com (C.H.); 15762181963@163.com (Q.S.); hebut_2017@126.com (Z.N.); 18822029985@163.com (G.Y.); wzying33@163.com (Z.W.)

**Keywords:** iPP, superhydrophobic microporous membrane, micro-molding, TIPS, membrane distillation

## Abstract

A facile method combining micro-molding with thermally-induced phase separation (TIPS) to prepare superhydrophobic isotacticpolypropylene (iPP) microporous membranes with micron-submicron hierarchical structures is proposed in this paper. In this study, the hydrophobicity of the membrane was controlled by changing the size of micro-structures on the micro-structured mold and the temperature of the cooling bath. The best superhydrophobicity was achieved with a high water contact angle (WCA) of 161° and roll-off angle of 2°, which was similar to the lotus effect. The permeability of the membrane was greatly improved and the mechanical properties were maintained. The membrane prepared by the new method and subjected to 60h vacuum membrane distillation (VMD) was compared with a conventional iPP membrane prepared via the TIPS process. The flux of the former membrane was 31.2 kg/m^2^·h, and salt rejection was always higher than 99.95%, which was obviously higher than that of the latter membrane. The deposition of surface fouling on the former membrane was less and loose, and that of the latter membrane was greater and steady, which was attributed to the micron-submicron hierarchical structure of the former and the single submicron-structure of the latter. Additionally, the new method is expected to become a feasible and economical method for producing an ideal membrane for membrane distillation (MD) on a large scale.

## 1. Introduction

Membrane distillation (MD) is a thermally driven process that uses hydrophobic porous membranes as the barrier. The driving force lies in the vapor pressure difference between porous hydrophobic membrane surfaces, which allows the volatile components in the solution to permeate to another side [1,2,3]. Compared with the conventional evaporation process, MD has advantages, such as equipment simplicity, a low expense, a mild operating temperature, and a low operating pressure, and it is receiving an increasing amount of attention in the fields of wastewater treatment, seawater desalination, pharmaceutical processing, etc. [4,5]. Several conventional and commercial hydrophobic materials, such as polyvinylidenefluoride (PVDF), polytetrafluoroethylene (PTFE), and isotacticpolypropylene (iPP), are used to prepare MD-related membranes [6,7]. However, pore wetting and membrane fouling easily occur when the hydrophobicity of the membrane is insufficient, which causes failure of the membrane distillation process, and consequently degrades the salt rejection [8,9]. Therefore, it is a continuing challenge to fabricate MD membranes with anti-wetting or superhydrophobic characteristics [10].

Inspired by self-cleaning phenomena in nature, researchers have developed several methods to fabricate superhydrophobic porous membranes with hierarchical structures [11,12]. The methods are mainly divided into two categories: the roughening modification of existing hydrophobic membranes and the direct preparation of membranes with hierarchical structures via the phase separation method [13].

The first strategy includes grafting [14,15], spraying [16,17], plasma etching [18,19], etc. Generally, the coating on the membrane surface prepared by grafting or spraying increases the mass transfer resistance, and leads to a decline in flux. The adhesion between the coating and the membrane surface deteriorates after a long work period [20]. In addition, plasma etching destroys the mechanical properties of the membrane and always costs a lot, which leads to its limited application [7,21].

Compared with the first strategy, the second strategy has the advantage of including a simpler and more economical preparation process [22,23]. Therefore, the method of preparing superhydrophobic microporous membranes by phase separation has become mainstream [24]. The strategy constructs the hierarchical structure by adjusting the phase separation conditions, such as adding inert additives into the casting solution or coagulation bath [25] and using vapor induction [26]. However, the mechanical stability of the surface structures obtained by these methods is unsatisfactory. In addition, extensional phase separation methods have been proposed, such as dilute solution coating [27], blending nano-particles [28], and adhering nano-particles to the membrane surface [29]. However, the uniformity of the dilute solution on the membrane surface is difficult to control. The adhesion between nano-particles and the membrane surface is also unsatisfactory [30,31].

Unlike these methods, Wessling et al. [32] proposed a unique method called phase separation micro-molding (PSμM). The polymer solution was cast into a microgrooved mold and then immersed in a coagulation bath. The nascent membrane was peeled away from the mold, and the replicated membrane surface was obtained. Because the micro-molding process did not change the body structure of the membrane, the mechanical properties of the membrane were not affected. However, the size of the silicon wafers prepared by photolithography combined with deep reactive ion etching was limited. Inspired by PSμM, our group [33,34,35] proposed making highly hydrophobic or superhydrophobic PVDF microporous membranes by combining micro-molding with non-solvent-induced phase separation (NIPS) (the method was abbreviated as micro-molding & NIPS). The mold was extended from micro-patterned silicon wafers to the mesh embedded in polydimethylsiloxane (PDMS), water-resistant sandpaper with a certain roughness, and an aluminium metal mold that corroded in the specific acidic aqueous solution. In this method, the homogeneous polymer solutions were cast on the micro-structured mold and solidified via the NIPS process. The nascent membrane was peeled away from the mold, and a microporous membrane with a hierarchical structure was obtained. Similar work to that presented by this group has also been proposed by other scholars [36,37], but the above method is only applicable to polymers that dissolve at room temperature, such as polyvinylidenefluoride (PVDF), polyether sulphone (PES), or polyacrylonitrile (PAN). For insoluble polymer materials at room temperature, such as isotactic polypropylene (iPP), polyethylene (PE), or poly-1-butene (iPB), which are also available to prepare membranes, the method is inapplicable.

Unlike NIPS, thermally-induced phase separation (TIPS) is suitable for various polymer membrane materials, including most polymer materials that are insoluble in solvents at room temperature and cannot be formed by NIPS. Therefore, the method of micro-molding with thermally-induced phase separation (micro-molding & TIPS) is proposed. It can be expected that this method could break through the limitations of micro-molding and NIPS and broaden the selection range of membrane materials. Compared with the above hydrophobic materials, iPP has economic advantages and a better chemical resistance, so iPP was selected as the membrane material in this study.

In this study, the composition of the polymer solution was fixed, and the effects of the micron-structure scale on the micro-structured mold and temperature of the cooling bath on the membrane structure and performances were investigated. The resulting membranes were systematically characterized by scanning electron microscopy (SEM), atomic force microscopy (AFM), the water contact angle (WCA), membrane pore sizes, N2 flux, and mechanical property measurements. Finally, the desalination performance and anti-fouling properties of the resulting membranes were measured by a vacuum membrane distillation (VMD) experiment. More importantly, the anti-fouling properties of the resulting membranes were further studied.

## 2. Materials and Methods

### 2.1. Materials

Smooth aluminium plates (150 mm × 120 mm × 2 mm) (SAP) were supplied by Tianjin Taihengtong Steel Company Co., Ltd. (Beichen District, Tianjin, China). Isotactic polypropylene (iPP) (T30S, melt index 3.06 g/10 min) was purchased from Petrol China Daqing Petrochemical Company Co., Ltd. (Longfeng District, Daqing, China). Di (2-ethyl-hexyl)-phthalate (DOP, AR), di-n-butyl-phthalate (DBP, 99.5%), ethyl ethanol (99.5%, AR), and inorganic salts (HClO and Fe_3_O_4_) were procured from Tianjin Bodi Chemical Co., Ltd. (Hexi District, Tianjin, China). Deionized water was produced by the marine technology centre of HEBUT. All chemicals were used without further purification in this work.

### 2.2. Experimental Methods

#### 2.2.1. Fabrication of Micro-Structured Molds

The micro-structured molds were fabricated by sandblasting. The sand used was brown fused alumina quartz sand. The sandblast pressure was set to 0.6 MPa. Before the sandblasting process, the nozzle stayed in the “empty blasted” mode for 3–4 min, to remove the moisture in the pipe used to transport quartz sand. Quartz sand was evenly blasted onto different plates with the nozzle inclined at 30°–40° and slowly moved back and forth. The sandblast time was set to 30 s. After the sandblasting process, the floating sand attached to the surface of the aluminium plate was purged with an air gun, and then washed clean to obtain the micro-structure mold with hill-shaped bulges and valley-shaped pits. Here, sand particles with two average sizes of 220 meshes (approximately 68 μm) and 320 meshes (approximately 45 μm) were selected for the preparation of the 220# micro-structured mold (220-MSM) and the 320# micro-structured mold (320-MSM), respectively.

#### 2.2.2. Preparation of the Membranes by the Micro-Molding & TIPS Method

First, iPP, DBP, and DOP were added to the flask at a ratio of 1:1.5:2.5 (wt: wt: wt) and heated at 200 °C under the protection of nitrogen until a homogeneous solution was achieved. Then, the polymer solution was degassed to remove gas present in the solution for use.

Figure 1 shows the preparation process of iPP membranes using the micro-molding & TIPS method. First, the casting solution was poured onto the micro-structured mold, which was preheated to 180 °C. Second, a self-made casting knife that was also preheated to 180 °C was used to evenly cast the solution onto the micro-structured mold. A liquid membrane of about 200 μm was obtained. Then, the mold with the liquid membrane was immersed in a cooling water bath with a certain temperature to complete the TIPS process. Subsequently, the nascent membrane was slowly peeled away from the mold. The obtained nascent membrane was first soaked in ethyl acetate for 48 h, and then soaked in ethanol for 24 h, to extract the residual organic solvents. Finally, the wet membranes were dried at room temperature. The above process was the complete micro-molding & TIPS process. Since the phase separation process was performed on the micro-structured mold, the submicron structures that originated from the TIPS process were grown on the replicated micron-structures, so the micron-submicron hierarchical structures were formed on the membrane’s bottom surface. The micron-submicron hierarchical structures of interest in this study were at the bottom surface of the membrane. For convenience, the surface in the context of this paper refers to the bottom surface of the membrane. The preparation conditions of different membranes are shown in Table 1.

#### 2.2.3. VMD Experiment

To verify the desalination performance and anti-fouling properties of the membrane, the VMD experiment was run for five continuous cycles. A cycle included ten hours of continuous VMD operation and half an hour for a cleaning process. Figure 2 shows a schematic diagram of the VMD experimental setup. There was a helical diversion groove in the membrane module, which was employed to realize cross-flow filtration. The effective area of each tested membrane was 11.68 cm^2^. Valves 1, 2, and 3 were open when the VMD operation was running. The preheated feed solution was driven into the membrane module by a magnetically driven pump and returned to the feed tank to form a cycle. The vacuum pump was used to generate a negative pressure at the permeate side (96 KPa). The vapor that was produced was passed through the membrane pores under a negative pressure, became condensed, and was then collected in a cold trap. A certain amount of water was added to the feed tank every 15–20 min to maintain a constant feed concentration during the experiment. The permeation flux (J) through the membrane and salt rejection during membrane distillation were calculated by the following equations [38]:(1)$$J=\frac{W}{A\cdot \tau }$$
(2)$$R=1-\frac{C_{2}}{C_{1}}$$
where W (kg) is the permeate water mass at 0.5h. A (m^2^) is the effective membrane area, $C_{1}$ is the conductivity of the feed solution, and $C_{2}$ is the conductivity of the permeate water. The electrical conductivity was measured by a DDS-307 conductivity meter (Shanghai Lida Instrument Plant, China).

During the cleaning process, the vacuum pump was stopped; valves 4 and 5 were opened; and valves 1, 2, and 3 were closed. Pure water was flowed through the membrane surface by a magnetically driven pump and turned to the collector. The membrane surface was cleaned.

During the entire VMD process, the temperature of the feed solution was set to approximately 70 °C, and the transmembrane pressure and feed solution (stimulated water) flow rate were 95 KPa and 30 L/h, respectively.

Seawater was taken from the Bohai Sea (Tanggu District, Tianjin, China). Pretreatment of seawater must be carried out before a membrane distillation operation. In total, 60 mg Fe_4_O_3_ and 8 mg HClO were added to the beaker filled with 2 L of seawater at 25 °C. The solution was stirred well. After 24 h, the supernatant was taken out and filtered, and the seawater for the VMD experiment was obtained. Detailed characteristics of the pretreated seawater are shown in Table 2.

### 2.3. Characterization

#### 2.3.1. Scanning Electron Microscopy (SEM)

The surface and cross-section morphology of the resulting membranes were observed by SEM (XL30, Philips Co., Amsterdam, The Netherlands). The cross-section fractured in liquid nitrogen and the membrane surface were sputtered with gold and then transferred to the microscope for imaging.

#### 2.3.2. Atomic Force Microscopy (AFM)

The surface morphology and roughness of the resulting membranes were measured using AFM (A5500, Agilent Co., Palo Alto, CA, USA). All membrane samples were imaged using a scan size of 25 μm × 25 μm and measured using the same tip. Finally, the roughness was obtained using the same tapping mode.

#### 2.3.3. Water Contact Angle (WCA) and Rolling-Off Angle

The wettability of the resulting membranes was measured via contact angle equipment (DSA-100, Kruss, Germany). First, 5 uL of water was dropped onto the resulting membrane to measure the contact angle (CA) value. A water droplet was dropped onto the resulting membrane at a certain angle to ensure that the water droplet could successfully roll to observe the roll-off angle. The tilt angle was also recorded at this time. The final results were averaged from more than five measurements.

#### 2.3.4. Specific Surface Area

The specific surface area was measured by SSA-6000 (BEIJING BIAODE ELECTRONIC TECHNOLOGY CO., Changping District, Beijing, China). The membrane samples were vacuumized at 80 °C and dried for 24 h. After drying, the static isothermal adsorption was measured at the liquid nitrogen saturation temperature (77 K). Relative pressures (P/P0) within the range of 0.05–0.995 and 52 pressure points were selected for the isothermal adsorption. The N_2_ adsorption curve of the sample was obtained by measuring the adsorption capacity of the sample at each relative pressure point. The specific surface area was obtained by linear regression with the Brunner−Emmet−Teller (BET) equation.

#### 2.3.5. Porosity, Pore Size Distribution, and N_2_ Flux

The porosity was determined by the dry-wet membrane gravimetric method. The resulting membrane (4 cm^2^) was weighed to obtain the mass of the dry membrane and soaked in dimethylbenze for 24 h; the mass of the wet membrane without dimethylbenze on the membrane surface was obtained. The porosity was calculated by the following equation [39]:(3)$$\varepsilon =\frac{W_{w}-W_{d}}{A\cdot \rho \cdot \delta }$$
where $W_{w}$ is the mass of the wet membrane (g), $W_{d}$ is the mass of the dry membrane (g), A is the area of the membrane (cm^2^), ρ is the dimethylbenze density (0.856 g/cm^3^), and δ is the thickness of the wet membrane (cm).

The pore size distribution was investigated by the bubble-point method. Nitrogen was forced to permeate from one side of the membrane to another side under a certain pressure. The relationship among the transmembrane pressure, the nitrogen flux through the dry membrane and the nitrogen flux through the wet membrane were obtained. The relationship between the pore size and the transmembrane pressure drop can be described by Equation (4) [40]:(4)$$d_{i}=\frac{0.63}{P_{i}}$$
where $d_{i}$ is the pore size and $P_{i}$ is the transmembrane pressure drop.

According to the Hagen–Poiseuille equation, the pore size distribution function based on the pore volume (V) can be expressed by Equation (5) [41]:(5)$$F_{v}=\frac{{{\bar{d}}_{i}}^{-2}\left[J_{i}/P_{i}-J_{i-1}/P_{i-1}\right]}{\left(d_{i-1}-d_{i}\right)\sum_{n-1}^{n}{{\bar{d}}_{i}}^{-2}\left[J_{i}/P_{i}-J_{i-1}/P_{i-1}\right]}$$
where $J_{i}$ is the nitrogen flux corresponding to $P_{i}$ and ${\bar{d}}_{i}$ is the average arithmetic value between $d_{i}$ and $d_{i-1}$. The mean pore size can be determined by the most probable value of the pore size distribution.

Here, the N_2_ flux through the dry membrane at 0.1 MPa was measured to characterize the permeability of the membrane. The average of three measurements was considered as the final result.

#### 2.3.6. Tensile Properties

The tensile properties of the resulting membranes were tested by a tensile tester (HD 021NS-5, Nantong Hongda experimental instrument Co, Nantong, China). Here, the stretching rate was set to 50 mm/min. The average of three measurements was considered as the final result.

## 3. Membrane Structure and Properties

### 3.1. Membrane Morphology

Figure 3 shows that there were many hill-shaped bulges on the 320-MSM surface (Figure 3b) with an average size of approximately 29.8 μm (the dimensions in this section were measured by Nano Measurer software), but there were no bulges on the SAP surface (Figure 3a). Therefore, sandblasting was an available method for preparing micro-structured molds with SAP. In addition, the size of the ordinary aluminium plates can be very large, so the size of the micro-structured mold was not restricted.

As shown in Figure 4a, the membrane obtained by the conventional TIPS was characterized as a mixed structure combining cellular pores and a crystalline particulate structure [40]. Among them, the submicron-scale cellular pore structure with an average size of approximately 0.72 μm originated from the liquid-liquid phase separation process, and submicron-scale crystalline particulate structures with an average size of approximately 0.63 μm originated from the polymer crystallization process. Compared with the cellular pores, the crystalline particulate structures were bulges, which contributed to the increase in hydrophobicity [33]. Therefore, the submicron-structures in this context refer to crystalline particulate structures.

As shown in Figure 4b_1_–e_1_, compared with the surface morphology of the membrane prepared by the conventional TIPS method (Figure 4a_1_), we noted that there were many hill-shaped bulges and valley-shaped pits on the membranes prepared by micro-molding & TIPS. On these micron-scale hill-shaped bulges and valley-shaped pits, there were many submicron-structures, which indicated that these hill-shaped structures and valley-shaped structures originated from the micro-molding process and the submicron-structures originated from the TIPS process. Hence, the micron-submicron hierarchical structures could be considered to have originated from the cooperation between the micro-molding process and the TIPS process. In the cross-section SEM images (Figure 4a_4_–e_4_), the membrane presented mixed structures combining cellular pores and particulate structures.

#### 3.1.1. Effects of the Mold Type

SEM images of the membranes prepared by 220-MSM and 320-MSM in identical phase separation conditions (t_bath_: 45 °C) are shown in Figure 4b_1_–b_4_,c_1_–c_4_. The submicron-structures that originated from TIPS had similar sizes. The cellular pore structure in the cross-section pictures was also similar, but the micron-structure of the membrane surface was obviously different. Here, the micron-structure of membrane b had a larger size (approximately 18.5μm), smaller degree of prominence, and smaller distribution density than those of the other membranes. In contrast, the surface of membrane c had a smaller size (approximately 10.39 μm), greater degree of prominence, and greater distribution density. The difference in prominence came from the sand blasting process. Smaller sand particles were easily embedded in the mold, and the embedding depth was very deep, which made the replicated micro-structures more prominent. Therefore, the surface of membrane c looked rougher than the surfaces of the other membranes.

To further explore the roughness of the membrane surface prepared by different micro-structured molds, AFM characterization of membrane a, membrane b, and membrane c was performed, as shown in Figure 5. As we know, Ra is defined as the average deviation of the valleys and peaks relative to the mean plane [9]. The result showed that the membrane prepared by SAP with an average roughness of 0.063 μm was very flat, while the Ra of the membrane surface obtained by 220-MSM and 320-MSM significantly increased, and was 0.3236 and 0.4407 μm, respectively. The hierarchical structures on the surface of membrane b were larger and less abundant than those on the other membranes, while those on the surface of membrane c were smaller and more abundant. The combination of these factors resulted in a rougher membrane being obtained by 320-MSM, which was consistent with SEM.

#### 3.1.2. Effects of the Cooling Bath Temperature

Figure 4c_1_–c_4_,d_1_–d_4_,e_1_–e_4_ depicts the SEM images of the resulting membranes prepared by 320-MSM at different t_bath_. In the surface SEM images, these membranes had approximately similar micron-structures, but the size of the submicron-structures gradually increased with increasing t_bath_ (*d*: approximately 0.47 μm, *c*: approximately 0.67 μm, and *e*: approximately 0.86 μm). In the enlarged cross-section SEM pictures, the cellular pores also became larger. With the increase in t_bath_, the growth time of the submicron-structures increased, so the size increased, which indicated that regulation of the TIPS conditions can be an effective method for controlling the size of the submicron-structures.

### 3.2. Hydrophobicity

#### 3.2.1. Effects of the Mold Type

Figure 6 (membranes a–c) describes the effect of the micron-structures of the mold on the hydrophobicity. Comparing membranes a, b, and c, the WCA gradually increased, and the roll-off angle showed the opposite trend. The WCA of membrane c increased to 161° and the roll-off angle decreased to 2°, which achieved superhydrophobicity. As described in Section 3.1, compared with membrane b, the micron-structures on membrane c had a smaller size, a greater degree of prominence, and a greater distribution density, which led to shorter distances. As shown in Figure 7a, when a water droplet was in contact with membrane b, the water droplet hung over the micro-bulges, and part of it was immersed in the micro-bulges, presenting a Cassie–Wenzel state, so the water contact angle was low [42]. As shown in Figure 8B, when a water droplet was in contact with membrane c, it was suspended entirely on the micro-bulges and presented a complete Cassie state, which exhibits superhydrophobicity [43].

#### 3.2.2. Effects of the Cooling Bath Temperature

Figure 6 (membranes d, c, e) demonstrates the effect of t_bath_ on hydrophobicity. Comparing membranes d, c, and e, the contact angle increased and then decreased with the increase in t_bath_, and the roll-off angle displayed the opposite trend. All membranes achieved superhydrophobicity, but the roll-off angle was different. By studying the SEM images in Figure 4, we noticed that the size of the submicron-structures changed from approximately 0.47 μm to approximately 0.67 μm, and then to approximately 0.86 μm, which meant that micron-structures were of great significance for the static contact, and the size of the submicron-structures at a suitable value was conducive to the realization of the lotus effect.

### 3.3. Porosity, Pore Size Distribution, and N_2_ Flux

Table 3 shows the specific surface area of the resulting membranes. It was found that the specific surface area increased gradually by comparing membrane a, membrane b, and membrane c, which demonstrated that the 320# mold has the best molding effect.

As for membrane d, membrane c, and membrane e, the specific surface area increased first and then decreased. The molds used were the same, and the micro-structures were similar. The difference in the specific surface area of the three was due to the difference in the TIPS structure. With the increase of t_bath_, the cooling rate decreased, the driving force of phase separation was reduced, and the time of undergoing phase separation increased. The coalescence tendency of cellular pores induced by the liquid-liquid(L-L) phase separation increased, which resulted in a decrease in the number of cellular pores and an increase in size. Additionally, the particulate structures grew gradually, leading to a shorter gap between them. Therefore, the internal connectivity of the membrane body structure became better. The combination of the two promoted the final change.

Figure 8A shows the porosity of the resulting membranes. Among them, membranes a, b, c show the effect of the mold type on the porosity, and membranes d, c, e show the effect of t_bath_ on the porosity. Comparing a, b, and c, there were no significant differences among the porosities of membrane a, membrane b, and membrane c. With the same TIPS, the enlarged cross-section SEM images in Figure 4a_4_–c_4_ showed that the body structures of the three membranes were similar. However, the surface structures were different, which demonstrated that the micro-molding had no effect on the porosity. By comparing membrane d, membrane c, and membrane e, it was shown that the porosity gradually decreased with increasing t_bath_. The time of undergoing the TIPS process was prolonged, and the particulate structures and cellular pore structures of the body structure increased, which led to a larger shrinkage tendency and decreased the supporting property. Therefore, the porosity decreased.

Figure 8B shows the pore size distributions of the resulting membranes. The range of pore size distributions of all the membranes was narrow; the maximum pore size of the resulting membranes was no more than 0.28μm, which was closely related to the TIPS process [43]. Based on the principle of the bubble-point method, the mean pore size was positively correlated with the permeability. Therefore, all factors contributing to the membrane permeability were helpful for improving the mean pore size. For the porous membrane, the permeability is affected by 1) the compactness of the membrane surface and the specific surface area of the membrane surface, and 2) the symmetry of the membrane body structure and the internal connectivity of the membrane body structure. Among them, the compactness of the membrane surface, the symmetry of the membrane body structure, and the internal connectivity of the body structure were controlled by the cooling rate [44].

Figure 8B (membranes a, b, c) describes the effect of the mold type on the pore size distribution. Membrane a, membrane b, membrane c were prepared by the same cooling rate, so the compactness of the membrane surface, the symmetry of the membrane body structure, and the internal connectivity of the membrane body structure of those membranes were similar. The difference between them was the surface morphology caused by the mold, which resulted in a large difference in the specific surface. Therefore, the difference in the permeability of the three membranes was derived from the specific surface area, and the difference in the mean pore size was due to the specific surface area.

Figure 8B (membranes c, d, e) describes the effects of the cooling bath temperature. The mean pore size increased with the increasing cooling temperature. With the increase of t_bath_, the coalescence tendency of the cellular pore induced by L-L phase separation increased, which resulted in a decrease in the number of cellular pores and an increase in size, so the symmetry of the membrane body structure improved. Additionally, as mentioned above for the specific surface area, the internal connectivity of the membrane body structure became better with the increase of t_bath_. Meanwhile, the increased gap between particulate structures meant that the compactness of the membrane surface became worse. Therefore, the total connectivity of the membrane became better. Furthermore, the specific surface area increased first and then decreased. The combination of these factors caused a gradual increase in permeability, and the mean pore size thus increased.

Figure 8C shows the N_2_ flux of the resulting membranes. The N_2_ flux showed a similar direction to the pore sizes.

### 3.4. Tensile Properties

Figure 9 (membranes a, b, c) demonstrates the effect of the mold type on the membrane tensile properties. From the flat SAP to the rougher 320-MSM, the resulting membranes exhibited a slight decrease in tensile properties, which can be explained as follows: the rougher mold had a rougher membrane surface, which corresponded to a more obvious stress concentration phenomenon.

Figure 9 (membranes c, d, e) demonstrates the effect of the cooling bath temperature on the tensile properties of the resulting membranes. The tensile properties decreased with increasing t_bath_, which can be explained by the fringed-micelle model [45,46]: the tensile properties depend on the number of frenum molecules. As t_bath_ increased, the size of submicron-structures (crystalline particulate structures and cellular pore structures) derived from the TIPS process increased. There were fewer frenum molecules between particles, which resulted in a decrease in the tensile properties.

## 4. Performance of Anti-Fouling in the VMD Process

### 4.1. Permeate Flux and Salt Rejection

Because hydrophobicity has an important effect on the desalination performance and anti-fouling properties of membrane a, membrane b, and membrane c, which have significantly different hydrophobicity values, this was employed for comparative research. The obtained results are shown in Figure 10, Figure 11 and Figure 12.

In the entire 100 h VMD process, the fluxes of membrane b and membrane c were higher than that of membrane a, and membrane c performed the best. The flux of membrane a decreased to a certain extent with an increasing operation time, but that of membrane c with micron-submicron hierarchical structures slightly decreased. In the first cycle, the final permeate flux of membrane a decreased to 90.2% of the initial flux, while the flux of membrane c maintained a constant value. As can be seen in Figure 11, in the fifth cycle, the initial permeate fluxes of membrane a and membrane c decreased to 89.0% and 99.8% of the first cycle initial flux, respectively. The final flux of membrane a and membrane c decreased to 78.3% and 98.2% of the first cycle initial flux, respectively.

As can be seen in Figure 12, after five continuous cycles, the salt rejection of all membranes was more than 99.6%. The salt rejection of membrane b and membrane c was always greater than 99.8%, while that of membrane a displayed some degree of decline and decreased to 99.6%.

In the first cycle, the flux of membrane a was significantly reduced, so we speculated that membrane a was polluted, and the fouling on membrane a caused a decrease in flux. The fouling also aggravated the wetting of the membrane, so the salt rejection was reduced. Unlike the above phenomenon, the flux of the membranes with micron-submicron hierarchical structures slightly declined, which indicated that the membrane with micron-submicron structures had a better anti-fouling property. From the perspective of the recovery of flux and salt rejection, the initial flux in the fifth cycle of membrane b and membrane c recovered well after being cleaned. However, the flux of membrane a only recovered well in the first two cycles. The initial flux in the fifth cycle of membrane c recovered to 99.6% of the first cycle initial flux after cleaning, while that of membrane a only recovered to 89.0%, which revealed that the existence of micron-submicron structures improved the recovery ability. Additionally, the changing trends of flux and salt rejection in the fifth cycle of membrane c were steadier than those of membrane a, which further indicated that membrane c had a better anti-fouling property.

### 4.2. SEM Images after the VMD Process

The SEM images of the surface and cross-section of the membranes at the end of the five cycles are shown in Figure 13. In the cross-section images, there was no fouling inside the three membranes, but fouling appeared on the membrane surface, which suggested that the pollution behavior belonged to surface deposition. The fouling on the membrane surface changed from greater and steady (membrane a) to less and loose (membrane b/membrane c). In particular, the polluted points on membrane c were difficult to discover.

From the perspective of the flux recovery rate of Section 4.1, membrane a in the first two cycles exhibited a very good recovery, but could not return to the original flux in the subsequent three cycles. In terms of the flux decay rate, the decay rate in the fifth cycle of membrane a was very fast. Therefore, the fouling at the end of the five cycles of membrane a included the fouling that occurred in the first four cycles, in addition to that produced in the fifth cycle. In contrast, the membrane with micron-submicron hierarchical micro-structures had a high recovery rate and a slow decay rate, which implied that membrane c had a good anti-fouling property.

### 4.3. Discussion

As shown in Figure 10 and Figure 11, the desalination performance of the conventional membrane and the newly developed membranes in the initial stage was satisfactory; the rejection rate was 99.99%, which indicated that the membrane prepared by the iPP material was suitable for an MD operation. However, the desalination performance and anti-fouling property were obviously different for the long work period, which may be attributed to the change in structures from single submicron structures to micro-submicron hierarchical structures. First, the existence of the micron-submicron hierarchical structures contributed to a larger specific surface area and higher flux.

Next, membrane fouling caused by concentration polarization inevitably occurred in the MD process. For membrane a with a flat surface, there was a tendency to form precipitations and crystals on the membrane surface. As the process proceeded, this tendency tended to intensify. Therefore, membrane fouling was formed. In the later cleaning process, it could not be recovered well, which was consistent with the previous flux and SEM images presented in Section 4.1 and Section 4.2.

Compared with membrane a, membrane b and membrane c had good anti-fouling properties, which can be explained by their micron-submicron hierarchical structures. First, the existence of the micron-submicron hierarchical structure increased the roughness of the membrane surface. An unstable flow state appeared when the feed solution crossed through the membrane surface. The state improved the turbulence of the feed solution, which produced a shearing effect to destroy the formation and development of the boundary layer and weakened the membrane fouling and concentration polarization. Therefore, its salt rejection always maintained a high level, which verified its high anti-fouling property.

Second, the high hydrophobicity brought about by the micron-submicron hierarchical structures played an important role. When the feed solution crossed through the membrane surface, these micron-submicron hierarchical structures easily captured a large amount of vapor, which generated vapor pockets. This was characterized by the Cassie–Baxter equation:$$\cos\theta_{c}=f_{1}\cos\theta_{1}-f_{2}$$
where *θ_c_* is the contact angle on the composite surface and *θ*_1_ is the intrinsic contact angle. For iPP, *θ_1_* = 104º, *f_1_* is the contact fraction between the hierarchical surface and feed solution, and *f_2_* is the fraction of vapor trapped between the hierarchical surface and the feed solution.

Here, the *f_2_* values of membrane b and membrane c were 0.737 and 0.928, respectively. The amount of vapor trapped increased with the increase in *f_2_*, which produced more vapor pockets between the feed solution and the membrane surface. This effect was similar to the lotus effect. More vapor pockets resulted in a smaller contact area between the feed solution and the membrane surface, which implied that the feed solution or fouling only contacts the prominent area of micron-submicron structures. The factor was accompanied by the washing effect of the feed solution in the process of VMD, so fouling was hardly deposited. Even if some fouling was deposited on the membrane surface during the VMD process, the feed solution easily rolled on the surface with the lotus effect. Therefore, the fouling was easily carried away by the feed solution.

Furthermore, the lotus effect still played an important role in the cleaning process. Because the fouling was less dominant and loose, the membrane surface was easily recovered to clean under the action of washing water. These three factors above all indicated that micron-submicron structures had a good anti-fouling property.

### 4.4. Performance Comparison with Other Reported MD Membranes

A distillation comparison of the literature and this study is presented in Table 4. Compared with the membranes in other literature, the membrane in this paper showed a comparable or even better performance. The permeate flux reached 32.1 kg/m^2^·h, which indicated that the membrane has a good permeability. The salt rejection was higher than 99.9%, since the newly developed membrane has an excellent anti-fouling ability. It was confirmed that this method has development potential in industrial applications.

## 5. Conclusions

In this study, superhydrophobic iPP microporous membranes were fabricated by a novel approach of micro-molding & TIPS. This newly developed process ensured that the membrane surface replicated the micro-structures from the micro-structured mold, and the submicron-structures generated from the TIPS process “grew on” the micro-structures. These two structures collaboratively constructed micron-submicron hierarchical structures. The hydrophobicity of the membrane could be controlled by adjusting the size of micron-structures and the size of the submicron-structures; the best result achieved superhydrophobicity with a high WCA of 161° and low roll-off angle of 2°, which was similar to the lotus effect. The permeability of the membrane was greatly improved and the mechanical properties were better maintained. The 100h vacuum membrane distillation of the membrane prepared by the new method was compared with that of the conventional iPP membrane via the TIPS process. The former membrane had an excellent desalination performance and anti-fouling property, which were attributed to the existence of micron-submicron hierarchical structures. Additionally, the membrane prepared by micro-molding & TIPS has no limitation in terms of size. Therefore, the new method is expected to become a feasible and economical candidate method for producing an ideal membrane for MD on a large scale.

## Figures and Tables

**Figure 1 polymers-12-00962-f001:**
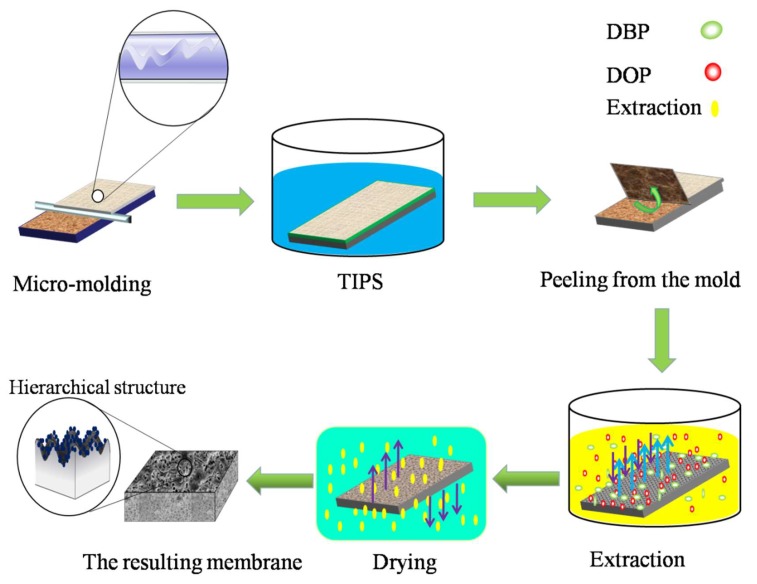
Schematic diagram of the micro-molding & thermally-induced phase separation (TIPS) method.

**Figure 2 polymers-12-00962-f002:**
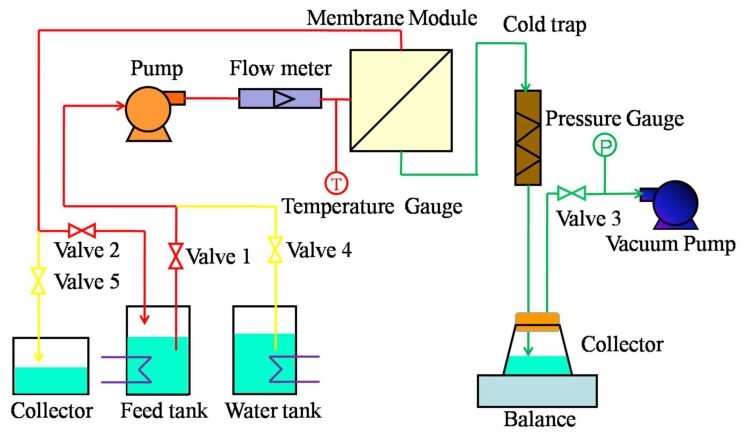
Schematic diagram of the vacuum membrane distillation (VMD) setup.

**Figure 3 polymers-12-00962-f003:**
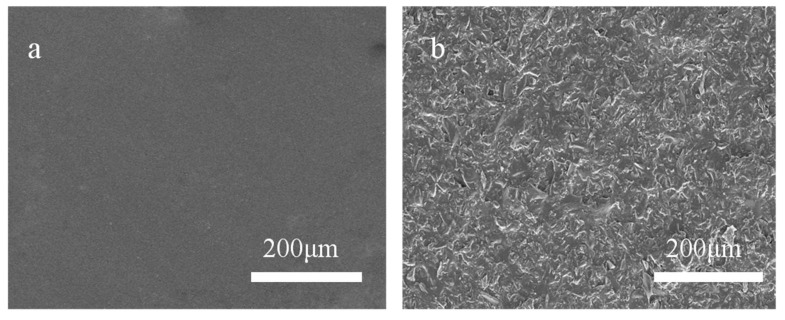
Scanning electron microscopy (SEM) images of the (**a**) SAP surface and (**b**) 320-MSM surface.

**Figure 4 polymers-12-00962-f004:**
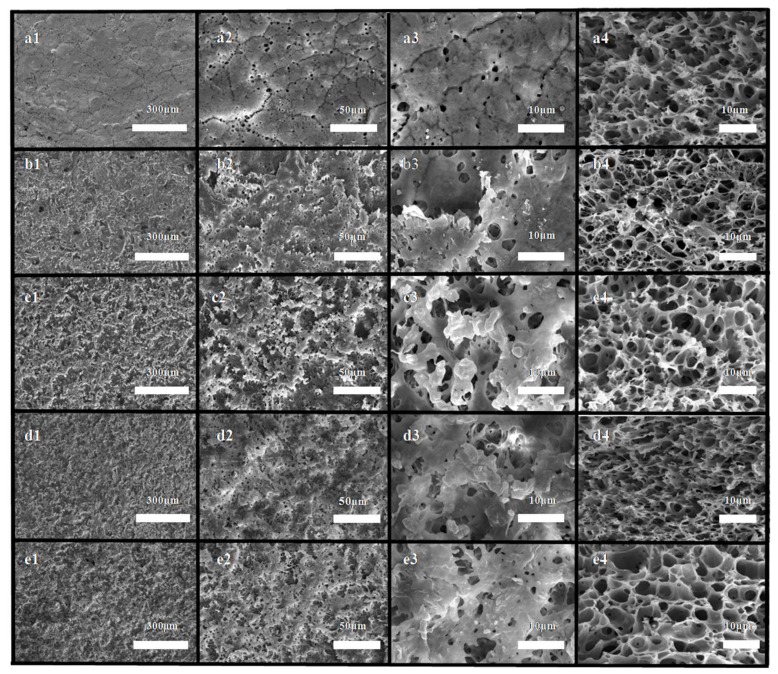
Images of the resulting isotacticpolypropylene (iPP) membranes. (**a_1_**,**a_2_**,**a_3_**,**a_4_**) t_bath_: 45 °C, SAP; (**b_1_**,**b_2_**,**b_3_**,**b_4_**) t_bath_: 45 °C, 220-MSM; (**c_1_**,**c_2_**,**c_3_**,**c_4_**) t_bath_: 45 °C, 320-MSM; (**d_1_**,**d_2_**,**d_3_**,**d_4_**) t_bath_: 40 °C, 320-MSM; and (**e_1_**,**e_2_**,**e_3_**,**e_4_**) t_bath_: 50 °C, 320-MSM. Surface morphology (1–3); cross-section morphology (4).

**Figure 5 polymers-12-00962-f005:**
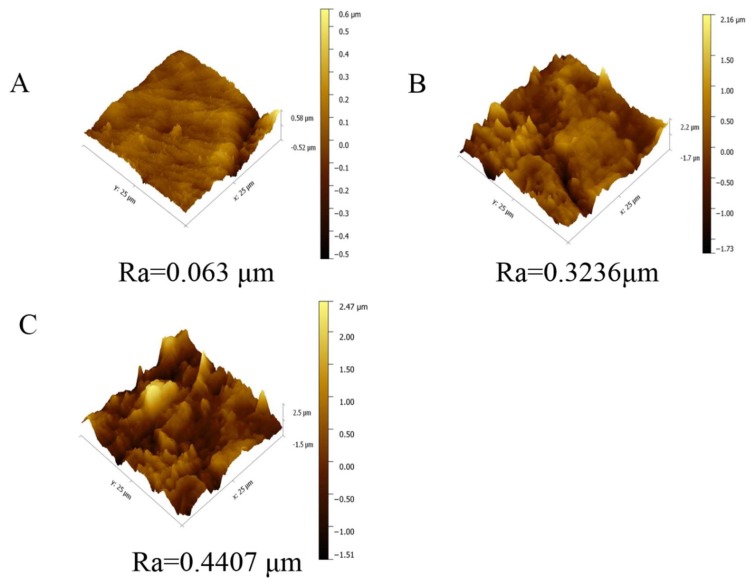
Atomic force microscopy (AFM) images of the resulting membranes prepared from different molds. (**A**) membrane a; (**B**) membrane b; (**C**)membrane c.

**Figure 6 polymers-12-00962-f006:**
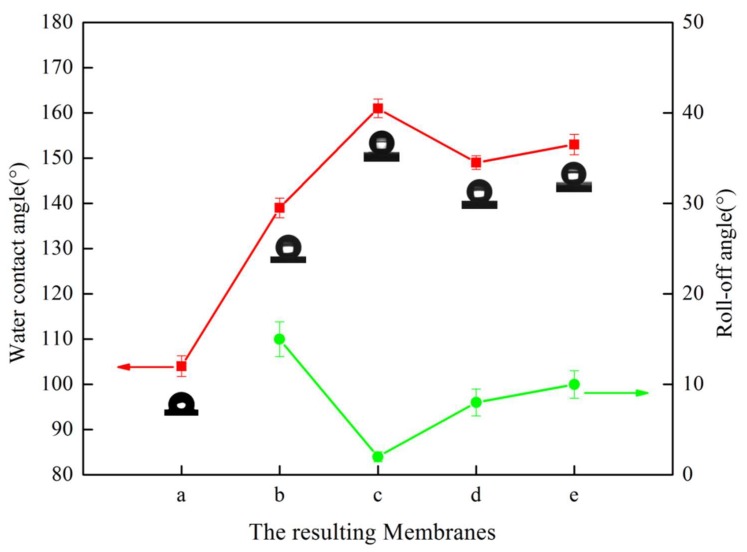
The water contact angle and roll-off angle of the resulting membranes. Videos of the roll-off angles are shown in the Appendix A; among them, the roll-off angle of membrane a was not measured due to high adhesion.

**Figure 7 polymers-12-00962-f007:**
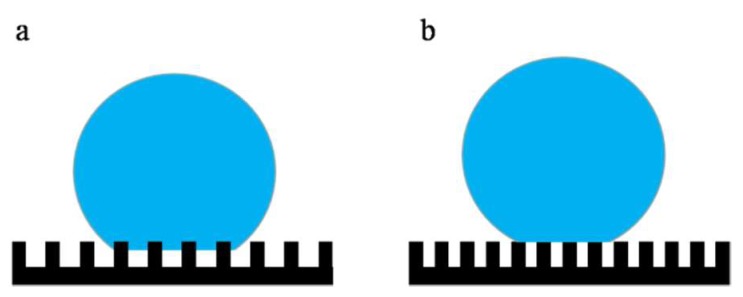
Schematic diagram of the Cassie–Wenzel model (**a**) and Cassie model (**b**).

**Figure 8 polymers-12-00962-f008:**
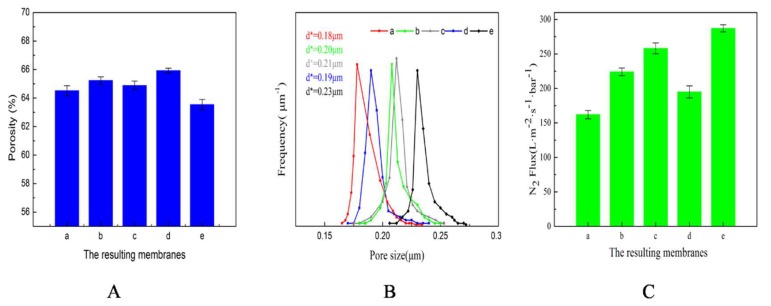
Porosity, pore size distribution, and N_2_ flux of the resulting membranes. (**A**) Porosity; (**B**) Pore size distribution; (**C**) N_2_ flux.

**Figure 9 polymers-12-00962-f009:**
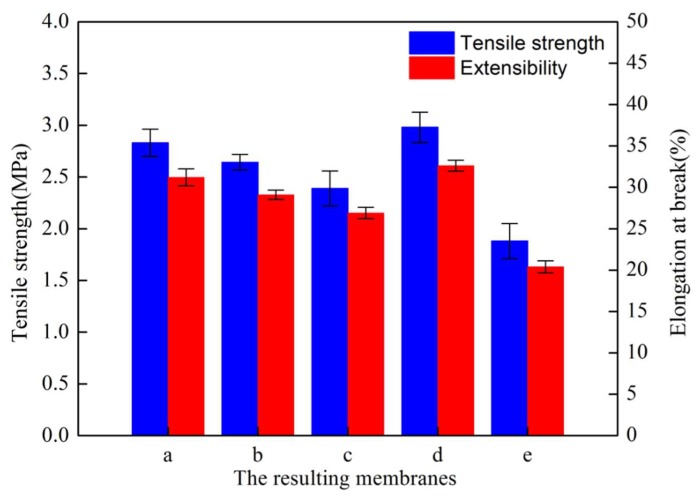
Tensile properties of the resulting membranes.

**Figure 10 polymers-12-00962-f010:**
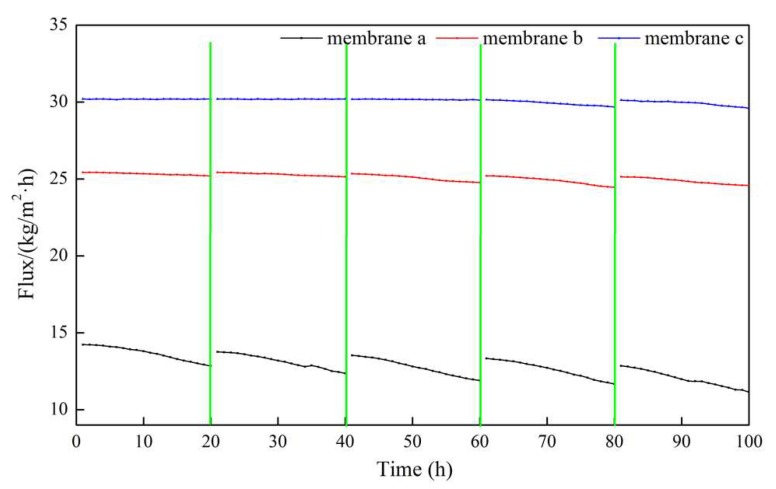
Dynamic permeate flux of membranes a, b, and c using actual seawater as the feed solution.

**Figure 11 polymers-12-00962-f011:**
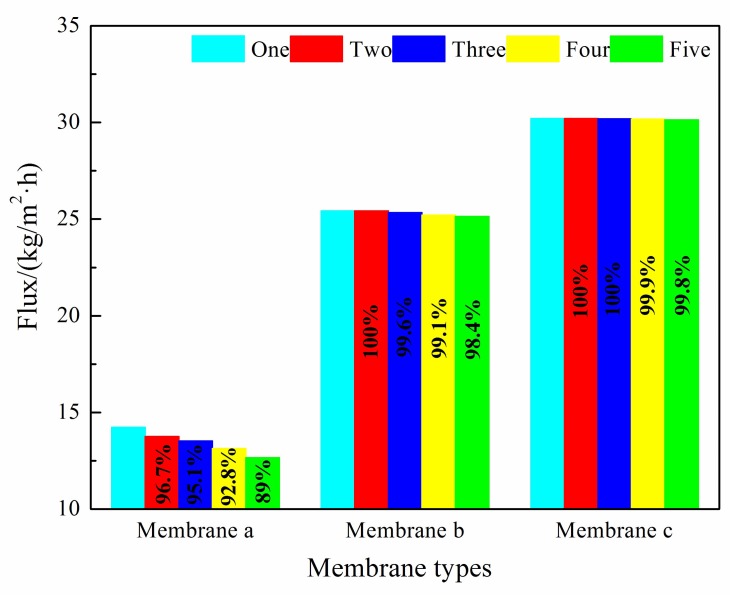
The maximum fluxes obtained for membranes a, b, and c.

**Figure 12 polymers-12-00962-f012:**
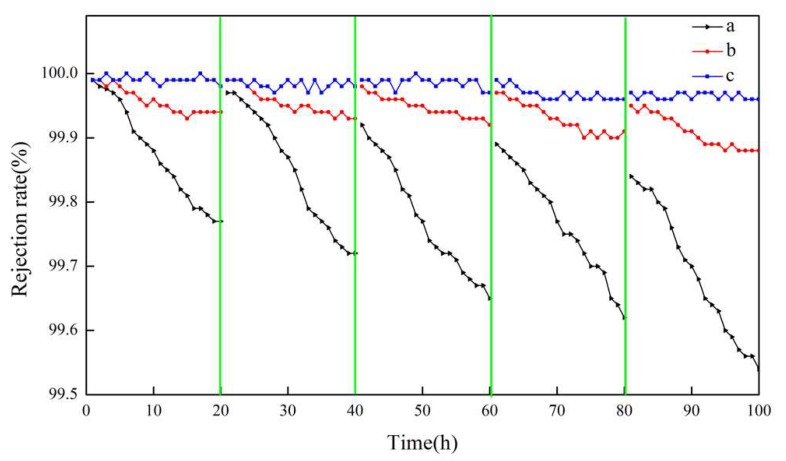
Salt rejection of membranes a, b, and c using actual seawater as the feed solution.

**Figure 13 polymers-12-00962-f013:**
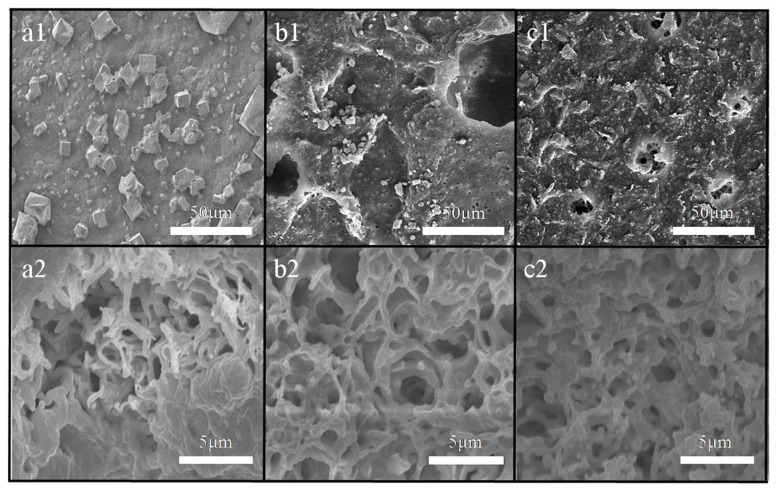
SEM images of the resulting membrane surface (**a1**,**b1**,**c1**) and membrane cross-section (**a2**,**b2**,**c2**) structure after a 100 h VMD experiment.

**Table 1 polymers-12-00962-t001:** Preparation conditions of the membrane (SAP: smooth aluminium plate; 220-MSM: 220# micro-structured mold; 320-MSM: 320# micro-structured mold).

Membrane	a	b	c	d	e
Mold type	SAP	220-MSM	320-MSM	320-MSM	320-MSM
Cooling bath temperature (°C)	45	45	45	40	50

**Table 2 polymers-12-00962-t002:** Characteristics of pretreated seawater.

Analysis Category	Measurement Value
PH	8.1
Conductivity (ms/cm)	45.2
Turbidity (NTU)	2.1
COD (mg/L)	0.9
Cl^−^ (mg/L)	17,210
Na^+^ (mg/L)	9541
Ca^2+^ (mg/L)	404.5
Mg^2+^ (mg/L)	1130.6
CO3^2−^ (mg/L)	51.2
SO4^2−^	2520

**Table 3 polymers-12-00962-t003:** Specific surface area of the resulting membranes.

Membrane	a	b	c	d	e
specific surface area (m^2^/g)	18.2	30.9	38.5	35.2	36.9

**Table 4 polymers-12-00962-t004:** Separation performance of several kinds of flat membrane for desalination.

Membrane	WCA (º)	Solution	Feed Flow (L/h)	T_f_ (°C)	Flux (kg/m^2^·h)	Rejection (%)	Stable Time (h)
PVDF/PFTs-SiO_2_ [47]	167.3	simulate seawater	180	76	25.2	99.9	13
PVDF [48]	155	simulate seawater	35	60	27.2	99.9	100
PVDF [49]	153	simulate seawater	24	70	31.6	99.9	8
PVDF-HFP [50]	162	seawater	18	70	34.1	99.9	250
PVDF-FDTs [51]	152.4	Seawater	192	70	2.78	99.9	120
This work	161	seawater	30	70	32.1	99.9	100

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
