# Peer review of "Facile Preparation of a Superhydrophobic iPP Microporous Membrane with Micron-Submicron Hierarchical Structures for Membrane Distillation"

_polymers, 2020, doi:10.3390/polym12040962_

Round 1
Reviewer 1 Report
The article deals with the preparation, by TIPS technique, of superhydrophobic microstructured iPP membranes by employing a micro-strucutured mold. Prepared membranes were characterized in terms of wettability, topography, morphology, porosity, pore size and mechanical resistance evaluating the effect of mold-type and coagulation bath temperature. The membranes were finally applied in VMD.
The article is very interesting for the journal and original. The work is well organized and well written in English. Many characterization tests have been carried out and properly discussed. The results are supported by the discussion. The referee suggests to accept the manuscript in the current form.
Author Response
Comments: The article deals with the preparation, by TIPS technique, of superhydrophobic microstructured iPP membranes by employing a micro-strucutured mold. Prepared membranes were characterized in terms of wettability, topography, morphology, porosity, pore size and mechanical resistance evaluating the effect of mold-type and coagulation bath temperature. The membranes were finally applied in VMD.
The article is very interesting for the journal and original. The work is well organized and well written in English. Many characterization tests have been carried out and properly discussed. The results are supported by the discussion. The referee suggests to accept the manuscript in the current form.
Response: Thank you for your positive comments on our work.

Reviewer 2 Report
This article presents the results of the preparation, membrane characterization and the application of the novel superhydrophobic iPP microporous membrane with micron submicron hierarchical structures for membrane distillation. The paper addresses the top theme of water desalination by using membrane techniques. Despite the fact that there are a lot of references to this topic, the results presented in this article are very promising and noteworthy. The authors present the new technique for the preparation of membranes, giving better results than for a conventional iPP membrane prepared via the TIPS process. Furthermore, a number of measurements were taken to characterise the structure of the resulting membranes and their properties. In general, the article is written well, it gives many details related to the examination of the membranes themselves and the process carried out. I recommended to accept the paper after having responded to the following comments:
2.2 Experimental methods
In the section Fabrication of micro-structured molds and Preparation of the membranes by the micro-molding &TIPS method the more details should be added.
- Performance of anti-fouling in the VMD process
The graph for the maximum fluxes obtained for membrane a, b, c, d should be added.
4.4 Performance comparison with other reported MD membranes
Please make more comparisons with other membranes used for desalination. I would suggest making a comparison with membranes tested on sea water, made with similar techniques and operating at the same temperature. Please make also a comparison graph for the analysed membranes.
